# The Moderating Effect of Corporate Governance on Corporate Social Responsibility and Information Asymmetry: An Empirical Study of Chinese Listed Companies

**Fahd Alduais** *[ID], **Nashat Ali Almasria** *[ID] **and Rana Airout**

Department of Accounting, Philadelphia University, Jarash Road, 20 KM, Amman 19392, Jordan
* Correspondence: falduais@philadelphia.edu.jo or accofahd@hotmail.com (F.A.);
  nalmasria@philadelphia.edu.jo (N.A.A.)

**Abstract:** This study is conducted to investigate the relationship between corporate social responsibility (CSR) and information asymmetry (IA), as well as the role of corporate governance (CG) as a moderating factor. This paper employs panel data regression analysis. The CSR disclosure scores are collected from the HX database by way of Hexun.com, while financial data are collected from the CSMAR database. The association between CSR and information asymmetry is examined using generalised least squares (GLS). The current evidence shows that CSR disclosure reduces information asymmetry. In addition, the findings illustrate that particular aspects of CG moderate the relationship between CSR and information asymmetry. More specifically, board size, CEO duality, and board independence positively affect the bid–ask spread. Moderation by the independence board positively affects the relationship between CSR disclosure and information asymmetry. Since the sample is derived from large Chinese companies, the results should be supported by samples obtained from the COVID-19 pandemic in 2020 and, subsequently, comparisons with the entire stock market. In future studies, we recommend conducting research using other variables as proxies regarding information asymmetry. The current study extends existing research on CSR and IA by adding both board characteristics and ownership concentration variables as moderating variables.

**Keywords:** corporate social responsibility; information asymmetry; corporate governance; disclosure; China

## 1. Introduction

CSR is a significant issue; therefore, CG has gained considerable attention in the last decade, owing to its role in demonstrating a company's responsibility and ensuring public oversight. Companies around the world, particularly those in China, engage in the practice of CSR. Since 2008, Chinese corporations have been required to publish their CSR initiatives. According to CSR research undertaken in China, ownership concentration, female leadership, members of the board with international experience, and political influence all have an impact on CSR performance (Yu and Chi 2021; Rehman et al. 2022; Yi et al. 2022; Cho et al. 2013; Wang and Li 2016; Xiang et al. 2021; Cui et al. 2018; Chang et al. 2012; Zu and Song 2009; Gu et al. 2013; McGuinness et al. 2017; Zhang et al. 2018; Lau et al. 2016). In addition to boosting transparency and performance, CSR decreases conflict between a corporation and its stockholders (Naqvi et al. 2021). Furthermore, CSR engagement is becoming increasingly common in business. The subject of whether CSR contributes to shareholder wealth remains fiercely debated. An additional investigation into corporate governance's impact on the connection between CSR and the involvement of asymmetric information might be conducted as part of this debate. The term "information asymmetry" refers to a dearth of information or knowledge among shareholders (Naqvi et al. 2021). Likewise, a number of members have a better understanding of market conditions in

contrast to others, and they also have different abilities and knowledge backgrounds with which to process information (Martínez-Ferrero et al. 2018).

The characteristics of board members, as well as ownership concentration, can affect the disclosure of CSR by firms. Shareholders and stakeholders will naturally encourage managers to engage more in CSR if it benefits shareholders. Although several prior studies have been conducted to investigate the effects of CSR on IA, they have encountered mixed results. Several authors believe that managers' self-interest drives CSR (Martínez-Ferrero and García-Sánchez 2015; Xiang et al. 2021). As a result, the shareholders' wealth suffers. According to separate research, CSR creates shareholder value by mitigating firm risk (Gong et al. 2018), reducing asymmetric information (Chowdhury et al. 2016; Hamrouni et al. 2022; Cui et al. 2018), enhancing the independence of the board (Fernández-Gago et al. 2016), building and promoting customer benefits (Rehman et al. 2022), and building moral capital among stakeholders (Lins et al. 2017). CSR practices in China are still in their early stages (Wang and Li 2016) and are frequently affected by political considerations (Chang et al. 2021; Yu and Chi 2021). Furthermore, because of their concentrated and state-dominated ownership, Chinese firms encounter acute agency conflicts (Jiang and Kim 2020; Yi et al. 2022; Chang et al. 2021).

According to this study, CG moderation has a considerable impact on CSR and IA among listed firms throughout China, which is the world's most rapidly growing industry and the largest emerging market in the world. On account of our research, we will be able to make three key contributions. Firstly, the current study adds CG factors as a moderating factor in CSR studies and IA (Cho et al. 2013; Cui et al. 2018; Lau et al. 2016; Hamrouni et al. 2022; Rehman et al. 2022). This paper specifically investigates how board features (size of the board, independence, and CEO duality) and shareholding influence the connection between CSR reporting and asymmetric information. These same results correspond to the results of a study performed by Xiang et al. in 2021, namely that executives are fairly likely to participate in CSR when they are given less interest and supervision by stockholders. This effect is particularly significant in organisations with substantial information asymmetry issues and is more likely to be a result of stakeholder distractibility—a greater ability to monitor incentive schemes. Owing to these findings, shareholders of Chinese listed companies would like to enhance corporate sustainability, implying that CSR benefits shareholders. These results are similar to those published by Wang and Li and Xiang et al. in 2016 and 2021, respectively, who report higher market valuations for CSR initiators than non-initiators, and Gong et al. and Xiang et al. in 2018 and 2021, respectively, who disclose that better CSR reporting lessens asymmetric information.

Secondly, this study also investigates whether social information disclosure directly reduces information asymmetry. Thirdly, to support the main conclusions, the present study contains annual trading value as a metric of asymmetric information to corroborate the primary findings. This proxy is employed by Draper and Paudyal (2008); Ness et al. (2001); Elbadry et al. (2015); Tessema (2019); and Linsmeier et al. (2002). Finally, the data indicate that CSR can significantly reduce the knowledge gap between management as well as external stakeholders. The study's conclusions have major consequences for investors and regulators, in addition to prospective studies on the subject. Companies are expected to share additional information as well as delivering meaningful financial information to the public. This provides each investor with a shared pool of information from which to decide to either purchase, sell, or hold a specific security. Individuals can only make effective financial decisions if they have constant access to reliable, substantial, and timely data.

The remainder of this paper is organised as follows: Section 2 reviews the literature and presents the development of our hypotheses. Section 3 describes the data and methodology. Section 4 presents the data analysis and results. Section 5 concludes the paper.

## 2. Literature Review and the Development of Hypotheses

### 2.1. CSR and Information Asymmetry

Arguments regarding CSR flourish without an obvious agreement on its significance. Evidently, CSR can be considered as an expansion of companies' endeavours to promote efficient CG, guaranteeing their sustainability by means of good corporate procedures that encourage responsibility and transparency to the whole community (Widyansyah et al. 2021). CG essentially implies a fair balancing of interests between corporate shareholders and different stockholders in a business (Nguyen et al. 2019; Al Maeeni et al. 2022; Almasria 2022b). Concerning discussions regarding whether neighbourhoods are major stakeholders, stakeholders are parties who can influence or are impacted by the planned outcomes of a company (Lu et al. 2021). On the stock exchange, investors are able to make decisions based on information that is vital for their decision-making processes (Shan et al. 2022; Naqvi et al. 2021). However, information asymmetry, which occurs whenever a participant in a transaction has more information than a competitor, is possible, resulting in market failure. Specifically, IA is a gap in knowledge and information (Diebecker and Sommer 2017; Naqvi et al. 2021). To conclude, IA is a situation in which one set of people appears to have greater access to confidential data relating to a company (Nguyen et al. 2019; Rehman et al. 2022; Naqvi et al. 2021). A variety of reasons can lead to CSR activities being undertaken. Due to the interest of market participants in learning more about CSR to make investment decisions, the current study emphasises CSR as a vital construct (Cho et al. 2013; Martínez-Ferrero and García-Sánchez 2015; Naqvi et al. 2021). The disclosure of CSR information by managers has increased in recent years in an attempt to increase openness (Dhaliwal et al. 2011; Cui et al. 2018), improve investor confidence and enhance market integrity (Hamrouni et al. 2022), reduce investment inefficiency, and, therefore, improve investment efficiency (Benlemlih and Bitar 2018). Consequently, CSR is gaining popularity among both academics and business leaders (Chang et al. 2021). Furthermore, asymmetric information refers to the increase in access to finances, which certain legislators have much more data on regarding market conditions, as well as different levels of abilities and knowledge in relation to interpreting data. Previous research has also shown that CSR activities improve a company's professional image (Greening and Turban 2000). Correspondingly, Lu et al. (2021) contend that CSR increases a company's profitability and creates a corporate image, which therefore enhances its reputation.

According to Hamrouni et al. (2022) and Cho et al. (2013), managers are significantly less likely to distort profit management by engaging in CSR activities, resulting in a narrower bid–ask spread (Dhaliwal et al. 2011; Cho et al. 2013; Nguyen et al. 2019). Research has been conducted to examine the effect of CSR clarity reporting on asymmetric information (Michaels and Grüning 2017; Liu et al. 2021; Hamrouni et al. 2022; Rehman et al. 2022; Nguyen et al. 2019; Diebecker and Sommer 2017; Naqvi et al. 2021; Cho et al. 2013; Cui et al. 2018). We hypothesise that organisations engaged in CSR or working to improve their CSR benefit from reduced information asymmetry in relation to investors, analysts, and management.

**H1.** *Corporate social responsibility is significantly negative with asymmetry information.*

### 2.2. CG and Asymmetry Information

Critical and ongoing discussions have focused on the relationship between CG and IA (Mohamed and Rashed 2021). Many studies, for example, have highlighted the importance of corporate governance initiatives in mitigating agency conflicts and asymmetric information (Abad et al. 2017). The goal of this research was to investigate the effects of various corporate governance practices on asymmetric information. Corporate governance studies consist of analysing board independence and size, CEO duality, and ownership concentration. Therefore, this study discussed four main hypotheses related to the relationship between a corporate governance mechanism and information asymmetry. According to the results of the study conducted by Chen et al. (2010), high levels of free cash flow and external funding are required to decrease asymmetric information among corporates

and enhance the influence and value of corporate governance procedures. CG refers to the existing rules and procedures that ensure management acts in the best interests of shareholders. CG is a tool for increasing shareholder value via organisational leadership (Almasria 2021). This has always been associated with agency problems and conflicts (Adel et al. 2019; Ullah et al. 2019; Gerged et al. 2018; Gerged and Agwili 2020). According to Hart (1995), CG mechanisms might arise as a result of managing and monitoring shareholders' or the board of directors' decisions on management tools and fiscal policies, which are indicated by debt leverage. This instrument governs a company's ownership structure, interactions with stakeholders, financial reporting, and information disclosure, along with the composition of the board of directors.

Furthermore, it helps create a particular skill that is required in strategic decisions and reduces information asymmetry issues (Feng et al. 2020; Yu and Wang 2018). Corporate governance systems influence the information disclosed by a company to its shareholders while also minimising the potential that management may act in its own interests. These strategies reduce the possibility that management is acting in its own interests by not fully disclosing relevant data to shareholders or providing less important data than necessary regarding reliability. Additionally, the efforts of management must be supported for managers to implement CSR in order to promote information transparency, eliminate information gaps, and reduce agency conflicts (Zhang et al. 2022; Cui et al. 2018; Naqvi et al. 2021). Furthermore, by coordinating with management, shareholders can use their voting rights to obtain multiple interests. Thus, concentrated ownership and the supervisory board are much more significant structures. According to our premise, strong CG reduces information asymmetry in the market, boosts market trading, and improves information transparency and disclosure.

### 2.2.1. Board Structure and Information Asymmetry

As a governance tool, the board of directors oversees managers to avoid conflicts of interest, which implies that the board of directors is accountable for financial as well as non-financial reporting transparency and integrity (Firmansyah et al. 2021; Dwekat et al. 2021). The board of directors is also in charge of safeguarding and preserving shareholders' interests (Almasria 2022a). Agency difficulties, according to agency theory, are created because agents and employers have competing interests (Jensen and Meckling 1976; Shleifer and Vishny 1997). As a conclusion, asymmetric information could be caused by conflicts of interest among several organisations (Hamrouni et al. 2022). Notwithstanding, many organisations are unable to measure and oversee managers' implementation behaviours and, therefore, agency charges are incurred (Haniffa and Hudaib 2006). Much more research concentrates on the relationship between corporate governance and asymmetric information, specific elements, or distinct corporate governance procedures (Rahayu et al. 2021). This signifies that the board of trustees does not allow management to influence the situation.

The size of the board of directors must be a decisive and influential factor in minimising management manipulations or inadequacies, as well as establishing boundaries regarding corporate disclosure and transparency. Researchers have discussed the ideal number of board members. Many claim that a small board promotes collaboration and appropriate decision making. Smaller boards of directors are less likely to experience challenges and disagreement from other members compared with larger boards of directors (Dimitropoulos and Asteriou 2010). Large boards, according to Hillman et al. (2007), encourage greater communication between companies and their surroundings. However, businesses can concentrate on stakeholder demands for CSR information and financial transparency. Consequently, there is the potential to monitor management and reduce information asymmetry among management and shareholders (Kanagaretnam et al. 2007). In contrast, as the board grows in size, it becomes less efficient at monitoring and increasing information asymmetry, which increases the volatility of trading value (Hamrouni et al. 2022; Tessema 2019). Board size (BS) appears to have an impact on the effectiveness of the board's control,

which in turn could have an impact on information communication and transmission (Hamrouni et al. 2022).

In terms of board independence, there is an indication that independent directors can improve analyst forecast accuracy (Goh et al. 2016). As a result, independent boards may be able to more closely oversee executive activity, reducing information asymmetry indirectly (Hillier and McClgan 2006; Holm and Schøler 2010; Elbadry et al. 2015; Abad et al. 2017; Tessema 2019; Hamrouni et al. 2022; Kanagaretnam et al. 2007) and supporting the accuracy of financial statements (Peasnell et al. 2005). Furthermore, past research, for instance (Core et al. 1999; Hermalin and Weisbach 1988), determined that board independence corresponds positively with firm value and poorly with financial fraud and earnings management (Klein 2002). Therefore, more efficient board independence improves the company's information quality and quantity, minimising information asymmetry. Previous research and theoretical studies indicated that board size reduces information asymmetry in the stock market and has a significant negative association with independent directors and information asymmetry. Thus, we propose the following hypothesis:

**H2a.** *Board size is significantly positive regarding information asymmetry.*

**H2b.** *With asymmetric information, the board's independence is substantially negative.*

CEO duality, which combines the CEO and chairman of the board, has been a source of debate in corporate governance circles. According to agency theory, CEO duality reduces the efficiency of the board's supervision, hence encouraging CEO entrenchment (Peng 2004). As a result, CEO dualism can have an effect on both business performance and reputation (Pham and Tran 2020). It can also influence what information a company shares, leading to suggestions that board diversity is significantly and negatively linked with voluntary disclosure. Executives' conduct can be influenced in two distinct ways: directly via incentives and indirectly via surveillance. The asymmetry of knowledge relating to the connection between their actions is lessened by an incentive that discloses the rewards and their amount of effort. Furthermore, information asymmetry can be mitigated via examples of legislative procedures, such as separating the chairman and CEO positions or boosting the number of non-board directors. It implies that ownership and control are not independent concepts. However, Kang et al. (2006) believe that the two-person CEO and capital structure has a good connection. They argue that the CEO's replication of the company decreases the agency's expenditure on separating ownership from control, thereby addressing the issue of information asymmetry. According to Abor (2007) and Fosberg (2004), whenever the chairman and the CEO differ from one another, a company's capital structure is more likely to represent the ideal corporate entity, implying further that the proportion of asymmetric information is reduced. Therefore, based previous research, we propose that CEO dualism benefits from information asymmetry and suggest the following hypothesis:

**H2c.** *CEO duality is significantly positively correlated with information asymmetry.*

2.2.2. Ownership Concentration and Information Asymmetry

Corporate governance prevents management from manipulating internal information, thereby reducing information asymmetry among investors, which ultimately reflects the robust performance of the stock price (Cho and Rui 2009; Core et al. 2008; Jiang et al. 2011). Corporate governance and financial market efficiency are inextricably linked. In a corporation with a high ownership concentration, a sole shareholder controls the vast bulk of the company's shares. Hence, by employing this principle, management can obtain an advantage over minority shareholders. According to Peng and Yang (2014) and Byun et al. (2011), in developing markets, the major shareholders already have the bulk of the shares and use private information to obtain an advantage and undermine the rights of smaller shareholders. This fosters a culture of information apathy among management and shareholders, resulting in an agency dilemma and increases in agency

expenses (Basuony et al. 2015; Jamalinesari and Soheili 2015; Bai et al. 2004; Salehi et al. 2014; Rehman et al. 2022; Jiang et al. 2011).

Management authoritarianism is associated with concentrated ownership according to Shleifer and Vishny (1997). Management authoritarianism in concentrated ownership, as shown by Fan and Wong (2002), reduces clarity and boosts agency costs because managers do not share knowledge with their competitors, resulting in increased agency costs and a deterioration in transparency. According to Perotti and von Thadden (2003), Pawlina and Renneboog (2005), Florackis and Ozkan (2009), and Shleifer and Vishny (1997), the majority shareholders could mitigate the risk of asymmetric information while also improving long-term efficiency. Higher levels of institutional ownership, on the other hand, were linked to a higher level of information asymmetry (O'Neill and Swisher 2003; Heflin and Shaw 2000; Fehle 2004).

**H$_{2d}$.** *Ownership concentration significantly and positively correlates with information asymmetry.*

### 2.3. The Moderating Influence of CG on CSR Disclosure with Asymmetric Information

The previous literature favours firm CSR reporting for several reasons (Xu et al. 2015; Wang et al. 2018; Marquis and Qian 2014; Jizi 2017; Cui et al. 2018; Yi et al. 2022; Hamrouni et al. 2022; Naqvi et al. 2021; Rehman et al. 2022; Adel et al. 2019). CSR reporting assists businesses in upholding their moral guidelines, minimising fraudulent conduct, lowering agency costs, and boosting efficiency (Lins et al. 2017). Companies that publish a high degree of CSR information are more likely to be approved for loan financing (Hamrouni et al. 2019). Although substantial research has been conducted on the effect of CSR reporting on asymmetric information (Cho et al. 2013; Cui et al. 2018), several scholars investigated the relationship between the board of directors' characteristics and asymmetric information (Hussain et al. 2018; Rutherford and Buchholtz 2007; Brennan et al. 2016; Cormier et al. 2010; Elbadry et al. 2015; Kanagaretnam et al. 2007; Abad et al. 2017). Only a handful of studies consider, as predictors of information asymmetries, both CSR disclosure and board characteristics (Cormier et al. 2011; Martínez-Ferrero et al. 2018; Hamrouni et al. 2022; Rehman et al. 2022).

According to agency theory, insiders will overinvest in corporate social responsibility so as to strengthen their reputations and establish themselves as socially responsible managers at the expense of shareholders (Buchanan et al. 2018; Yi et al. 2022). CSR investments, in essence, are an expensive deflection of a firm's precious resources as a consequence of shareholders' competing interests with management. According to McGuinness et al. (2017), equality for gender within senior management contributes to higher CSR performance. Board size, according to Hamrouni et al. (2022), is unrelated to CSR reporting with asymmetry information. Nevertheless, we anticipate that board size will have a bearing on the link between CSR compliance on asymmetric information. Furthermore, board independence will act as an effective intermediary in the relationship between CSR and IA. Furthermore, CEO duality will be crucial in managing connections between corporate social responsibility and asymmetry information. Furthermore, CSR with information asymmetry is likely to be moderated by ownership concentration.

**H$_{3a}$.** *CSR disclosure and information asymmetry, as mediated by board size, have a positive and significant relationship.*

**H$_{3b}$.** *CSR disclosure and information asymmetry have a positive relationship, which is mitigated by board independence.*

**H$_{3c}$.** *CSR disclosure and knowledge asymmetry, as mediated by CEO duality, have a positive and significant relationship.*

**H$_{3d}$.** *CSR and bid–ask spread mediation by ownership concentration have a positive and significant relationship.*

Figure 1 shows the theoretical framework of our paper, showing the different relationships between all the variables included in the study.

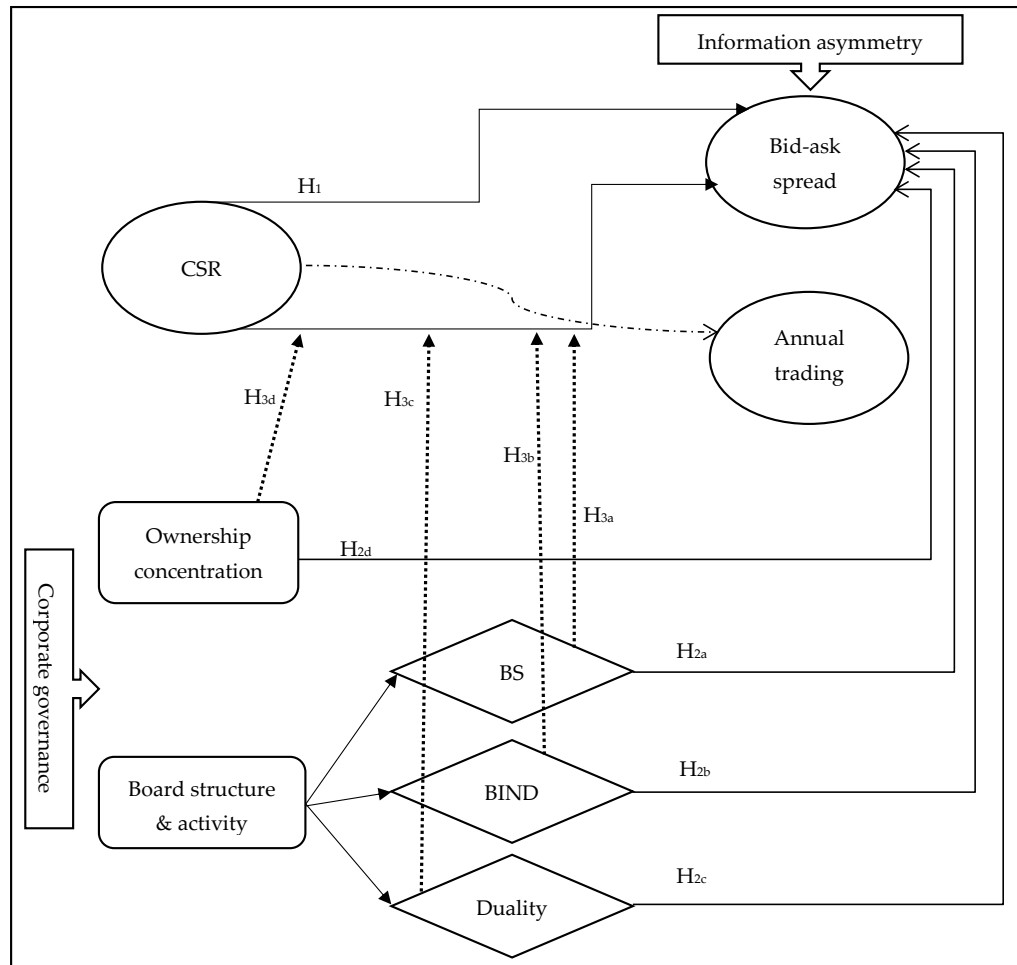

**Figure 1.** Theoretical framework.

## 3. Research Methodology

### 3.1. Sample and Data Collection

We included all CSMAR firms ranging from the years 2010 to 2019 listed on Chinese stock markets (Shenzhen and Shanghai), totalling 8358 Chinese businesses, in this study. With the exception of CSR data, which we obtained from Hexun.com (accessed on 15 March 2022), we employed data from the CSMAR database to deliver information on factors such as bid–ask, corporate governance, China Stock Market Trading, and China Stock Market Volume Trading. We excluded financial companies, such as banks and insurance firms, along with other diversified financial companies, due to significant variations in capital structures and regulatory environments, particularly compared with other businesses (Switzer and Wang 2013; La Porta et al. 2002; Jiang et al. 2011).

### 3.2. Measurements of Variables

#### 3.2.1. Dependent Variable

We used bid–ask spreads and annual trading value to calculate information asymmetry. Several studies have utilised bid–ask spread as an indicator of knowledge asymmetry (Flannery et al. 2004; Leuz and Verrecchia 2000; Kanagaretnam et al. 2005; Cheng et al. 2011; Rehman et al. 2022). Concerning bid–ask spread, if the shareholder is willing to pay lower than the owner is asking for, it is possible that the owner/seller and the potential buyer/shareholder have contradictory information (Glosten and Harris 1988; Welker 1995).

According to Cho et al. (2013), Kanagaretnam et al. (2005), and Cheng et al. (2011), a SPREAD is employed to calculate the proportion of each day's bid–ask spread to the closing price over a year (see Equation (1)). The bid–ask spread includes a liquidity component that safeguards investors against poor selection. Several academics concentrate on variations in bid–ask spreads in the aftermath of earnings announcements. This is in line with previous research on asymmetry information around similar incidents (Krinsky and Lee 1996; Libby et al. 2002; Kanagaretnam et al. 2007). We count the bid–ask spread

$$\%\text{Bid-ask SPREAD} = (\text{Ask} - \text{Bid}) \times 100/(\text{Ask} + \text{Bid}) \div 2 \tag{1}$$

where $\text{Bid}_t$ and $\text{Ask}_t$ are the daily bid and ask prices in year t, and CP is the daily closing price. We calculated IA by subtracting the daily ask price from the daily bid price and dividing by the closing price. We calculated the annual value of IA by averaging the entire daily IA readings. Following previous studies (Draper and Paudyal 2008; Ness et al. 2001; Elbadry et al. 2015; Tessema 2019; Linsmeier et al. 2002), we conducted a robustness check using trading value as a substitute for asymmetric information. Moreover, we examined how the moderated CG proxies impacted the associations between CSR and IA. Corporate governance procedures and different information asymmetry metrics support our findings. The trading value is the market worth of one of the shares traded for a corporation in a particular year. According to Acker et al. (2002), significant volumes of trade are related to closing prices that are typically on a daily basis range and indicate low levels of information asymmetry. A study by Gajewski (1999) concluded that trading volume was higher during days of declaration, indicating that the higher trading volume was due to the prospect of disclosing information. Accordingly, we anticipated that the greater the average annual trading value, the smaller the degree of information asymmetry.

### 3.2.2. Independent Variables

We developed a proxy for our sample companies' CSR engagement using CSR scores in accordance with earlier studies on CSR in China (Xiang et al. 2021; Yi et al. 2022; Gong et al. 2018; Marquis and Qian 2014; Lau et al. 2016; Zhang et al. 2018). HX uses a three-layer system to create CSR scores. Five factors are covered in the first layer: social responsibility, environmental responsibility, employee duty, shareholder responsibility, and customer responsibility. We included 13 criteria in the second layer, which supports the five dimensions, and included 37 criteria in the third layer, which supports the second layer.

### 3.2.3. Moderation Variables

We conducted a regression analysis of corporate governance in two stages. Firstly, we evaluated the effect of the variable directly on information asymmetry adopted from the literature (Abad et al. 2017; Tessema 2019; Ataullah et al. 2014; Coller and Yohn 1997; Holm and Schøler 2010; Alves et al. 2015; Chang et al. 2021). Subsequently, we examined its role as a mediator among information asymmetry and corporate social responsibility. The restriction variables grouped into three categories reflect the CG mechanism. Specifically, following previous studies, our choice comprises three categories that refer to board structure using independent directors (the number of independent directors divided by the total number of board members equals the proportion of independent director) and board size (the total number of directors on the board) as proxies for CG (Tessema 2019), board activity utilising CEO dichotomy (a dummy variable is 1 for firms whose chairman and general manager are the same person; otherwise, it is 0), as well as ownership concentration (the percentage owned by the largest shareholder). As a result, we anticipated that the lower the percentage of independent directors on the board of directors, the larger the trustees' exposure to the agency's dilemmas and the deterioration in the value of trading as well as volume trading. We chose a proxy of the board's effectiveness as the duality. Therefore, there might be an advantageous relationship among the ask–bid spread with CEO duality as well as an opposite association with trading value and volume. We adopted CEO duality from earlier studies (e.g., Abu Qa'dan and Suwaidan 2019; Kang

and Ausloos 2017; Larcker et al. 2011; Hong et al. 2021; Abor 2007; McGuinness et al. 2017; Pham and Tran 2020) and assessed it as a dummy variable, with a value of one assigned if the CEO is also the chairman of the board and a value of 0 otherwise. Ownership concentration is the proportion of the largest shareholder, which can be negative or positive with information asymmetry proxies. We expected an inverse related to the ask–bid spread and a positive related to trading volume and value. In light of the reviewed research, we hypothesised that boardroom interactions and CSR disclosure could function as replacements and/or supplement minimising information asymmetry. Many studies have explored the relationship between CSR disclosure and information asymmetry (Sun et al. 2014; Kim et al. 2014; Rehman et al. 2022; Nguyen et al. 2019; Michaels and Grüning 2017; Cho et al. 2013; Cui et al. 2018). To consider this interaction and determine whether certain boardroom characteristics, specifically BS, board independence, CEO duality, and ownership concentration are moderate, we used cross-variables (Rehman et al. 2022).

### 3.2.4. Control Variables Measures

We used multiple regression models to select several factors related to information asymmetry as control variables. Our study is based on previous research (Ferrell et al. 2016; Zhang et al. 2018; Xiang et al. 2021; Hamrouni et al. 2022; Naqvi et al. 2021) that controlled a series of related variables to the attributes of the firms. Among such variables is company size (SIZE), which is the natural logarithm of the company's total assets in a/one million renminbi. Book to Market (BTM) is the total shareholder equity divided by the total market value. Return on assets (ROA) is calculated by dividing net income by total assets, whereas leverage (LEV) is the total liabilities divided by total assets. Recent research indicated that the nature of insider trading is dependent on a company's characteristics, which in turn, determine the degree of information asymmetry between insiders and external users. As an example, smaller companies tend to have a more significant market reaction to insider trading for the reason that they are expected to have a "significant part of the relevant information" transferred to the market through their transactions. Large companies may face less asymmetric information because they are generally more mature, have time-tested practices and policies, and receive more focus from the market and regulators (Harris 1994). The assumption is that revenue companies will provide more details to connect shareholders to their outstanding performance (Wallace et al. 1994). An important variable is leverage, which demonstrates that high external financing from borrowers has low disparities in information. We also applied the ratio of market value to equity book value as a control.

### 3.3. Model Specification

We used the generalised least squares regression (GLS) approach for the following equation, which was previously accustomed to evaluating the association among corporate social responsibility disclosure and information asymmetry:

$$\text{BAspd}_{it} = f\ (\text{CSR}_{i,t} + \text{control variables}_{i,t}) + \varepsilon_{it} \tag{2}$$

We investigated the moderating influence of CG on the connection in the second stage and the relationship between CSR disclosure and asymmetric information. Thus, we made use of the following model:

$$\text{BAspd}_{it} = f\ (\text{CSR}_{i,t} + \sum \text{CG}_{i,t} + \sum \text{CSR*CG}_{i,t} + \sum \text{control variables}_{i,t}) + \varepsilon_{it} \tag{3}$$

Corporate governance (CG) has three distinct characteristics: board composition, activities, and ownership concentration, each of which is controlled by four variables: board size (BS), board independence (BIND), and CEO duality (DC), in addition to ownership concentration (OC).

## 4. Data Analysis and Results

### 4.1. Descriptive Statistics

Descriptive statistics for the factors studied are summarised in Table 1. The bid–ask spread index includes a mean value of −2.396 and a standard deviation of 5.911. As a result of this significant amount, it is possible that the sampled firms have a substantial degree of information asymmetry, and that there is sufficient diversity within each company to undertake a significant examination of the Chinese market. The CSR has a low value of −18.45 and a high value of 82.434. The mean CSR is −2.636, with an SD of 8.063. According to the findings, Chinese enterprises score poorly in the social dimension of CSR. The size of the organisations' boards has a mean of 23.149 members, 14.7% of whom are independent directors, whilst 19.4% have CEO duality. In terms of control factors, descriptive data suggest that enterprises have a reduced average profit margin (ROA) of 0.035 and considerable debt, with a leverage ratio of 53.2% on average. Finally, the mean value of the variable business size is 22.137.

**Table 1.** Descriptive statistics.

|  | N | Mean | Median | SD | Minimum | Maximum |
|---|---|---|---|---|---|---|
| BAspd | 8358 | −2.396 | −1.846 | 5.911 | −12.666 | 54.956 |
| CSR | 8358 | −2.636 | −2.191 | 8.063 | −18.45 | 82.434 |
| ATV | 8358 | 23.655 | 23.609 | 1.103 | 17.181 | 28.533 |
| BS | 8358 | 23.149 | 22 | 5.175 | 11 | 66 |
| BIND | 8358 | 0.147 | 0.143 | 0.022 | 0.032 | 0.273 |
| Duality | 8358 | 0.194 | 0 | 0.396 | 0 | 1 |
| OC | 8358 | 31.572 | 30.466 | 18.315 | 0.06 | 89.41 |
| SIZE | 8358 | 22.137 | 21.815 | 1.704 | 15.418 | 30.815 |
| BTM | 8358 | 770.155 | 535.709 | 1140.110 | −14,045.56 | 30,459.27 |
| LEV | 8358 | 0.532 | 0.489 | 1.525 | 0 | 96.959 |
| ROA | 8358 | 0.035 | 0.038 | 0.247 | −20.548 | 0.98 |

Note: CSR = corporate social responsibility. BAspd = bid–ask spread. ATV = annual trading value. BS = board size. BIND = proportion of independent directors. CEO Duality = when chairman and CEO are same person. OC = proportion of largest shareholder. SIZE = firm size. BTM = book to market. LEV = leverage. ROA = return on assets.

### 4.2. Correlation Matrix

Table 2 depicts the correlation matrix, which highlights the linear bivariate connections among the variables considered. The correlations between the factors are low to moderate. The large percentages of Pearson's correlation coefficients were also statistically significant, yet high enough to indicate the presence of a multicollinearity issue. The influence of the linear relationship on the outcomes was investigated by computing the variables of variance inflation factors (VIF) in each variable. As a consequence of this, multicollinearity was not really determined. Most of the connections are of a high quality.

**Table 2.** Correlations Matrix.

|  | BAspd | ATV | CSR | BS | BIND | Duality | OC | SIZE | BTM | LEV | ROA |
|---|---|---|---|---|---|---|---|---|---|---|---|
| BAspd | — | | | | | | | | | | |
| ATV | −0.128 *** | — | | | | | | | | | |
| CSR | 0.329 *** | −0.222 *** | — | | | | | | | | |
| BS | 0.023 * | 0.286 *** | −0.068 *** | — | | | | | | | |
| BIND | −0.001 | −0.085 *** | 0.021 | −0.381 *** | — | | | | | | |
| Duality | 0.063 *** | −0.131 *** | 0.074 *** | −0.168 *** | 0.034 ** | — | | | | | |
| OC | −0.031 ** | −0.011 | −0.001 | −0.005 | 0.006 | −0.041 *** | — | | | | |
| SIZE | 0.035 ** | 0.624 *** | −0.139 *** | 0.432 *** | −0.111 *** | −0.186 *** | 0.108 *** | — | | | |
| BTM | 0.161 *** | −0.222 *** | 0.084 *** | 0.126 *** | −0.01 | 0.001 | 0.164 *** | 0.278 *** | — | | |
| LEV | −0.116 *** | 0.227 *** | −0.112 *** | 0.242 *** | −0.065 *** | −0.18 *** | −0.028 * | 0.45 *** | −0.13 *** | — | |
| ROA | 0.128 *** | −0.023 * | 0.077 *** | −0.039 *** | −0.014 | 0.088 *** | 0.089 *** | −0.067 *** | 0.015 | −0.47 *** | — |

Note: CSR = corporate social responsibility. BAspd = bid–ask spread. ATV = annual trading value. BS = board size. BIND = proportion of independent directors. CEO Duality = when chairman and CEO are same person. OC = proportion of largest shareholder. SIZE = firm size. BTM = book to market. LEV = leverage. ROA = return on assets. Standard errors in parentheses. * $p < 0.05$, ** $p < 0.01$, *** $p < 0.001$.

### 4.3. Results and Discussion

According to the literature described above, our findings reveal that boardroom qualities can be used as substitutes or supplements in regard to CSR disclosure to alleviate knowledge asymmetry. Using cross-variables, we investigate whether board structure and ownership concentration modify the relationship with both CSR reporting and asymmetric information. We investigated how three types of corporate governance structures and corporate social responsibility interact to produce CSR disclosure and information asymmetry in this study (Oh et al. 2018; Hamrouni et al. 2022). Figure 2 depicts this.

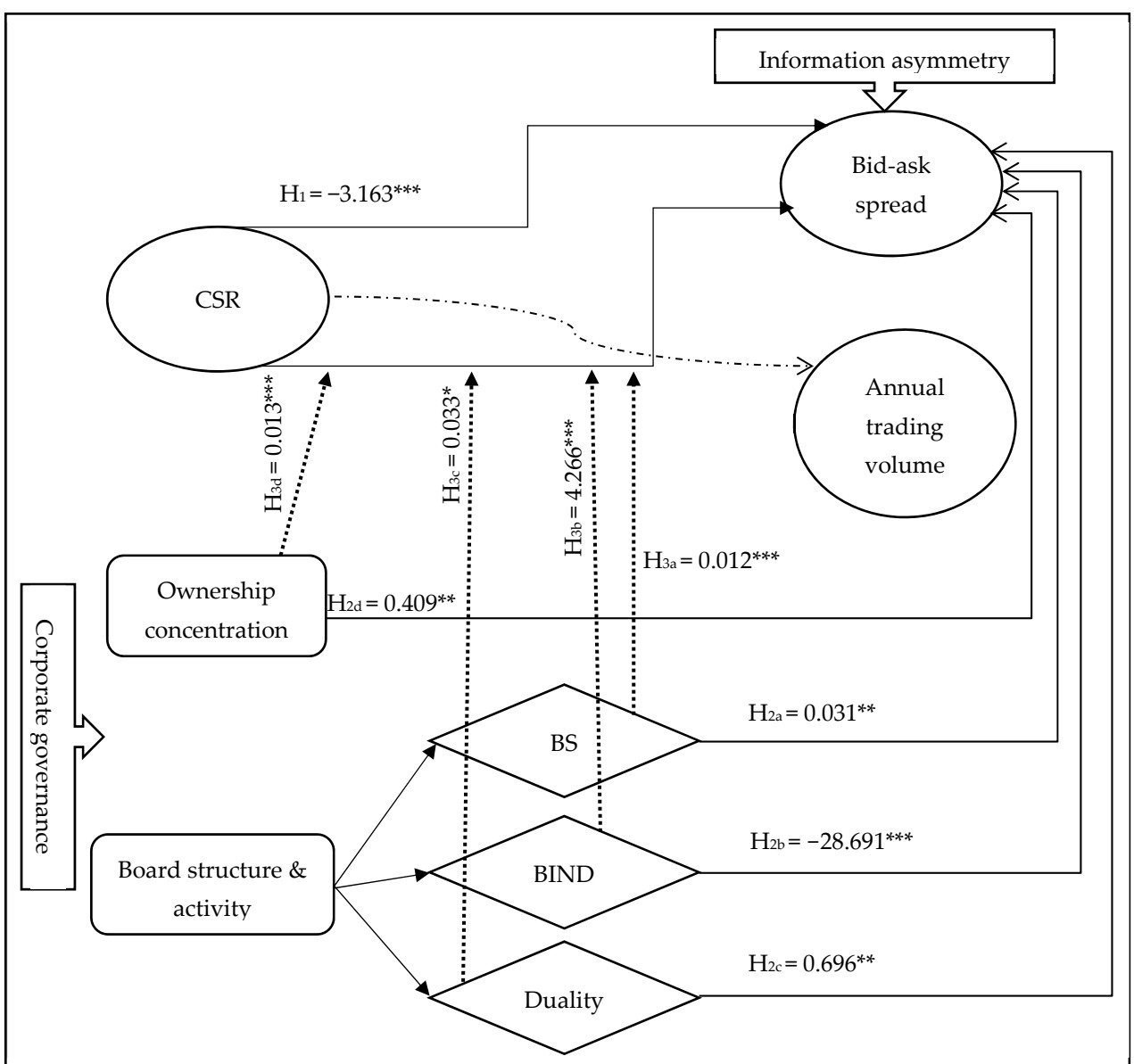

**Figure 2.** Summary of main regression analysis results. Note: * $p < 0.05$, ** $p < 0.01$, *** $p < 0.001$.

#### 4.3.1. Corporate Social Responsibility and Information Asymmetry

First, we explored the relationship between the social and information asymmetry indexes individually, with no moderating effects from board structure or owner concentration. The models appear to have a good fit. In fact, in Model 1, the coefficient explains the extent to which changes in the dependent variable can be explained by the independent variables (information asymmetry). In particular, the CG variables were separated into three categories (board structure, activity, and ownership concentration). CSR appears to

have a significant impact and negatively linked coefficient −3.163 *** at the 0.001 level according to Table 3. As a result, Chinese enterprises that disclose a substantial amount of social information have low information asymmetry. Therefore, our hypothesis **H₁** on the direct effect of CSR disclosure has been confirmed. In contrast to the literature (Magnanelli and Izzo 2017; Diebecker and Sommer 2017), we empirically demonstrated a positive association between CSR performance and information asymmetry. Furthermore, Hamrouni et al. (2022) established no relationship between corporate social responsibility and asymmetric information.

**Table 3.** Test of effect of CSR on information asymmetry.

| Dependent Variable—Information Asymmetry (Bid–ask Spread) | |
|---|---|
| **Predictor** | **Model (1)** |
| CSR | −3.163 *** |
| | (0.078) |
| BTM | 0.512 *** |
| | (0.109) |
| SIZE | 0.005 |
| | (0.937) |
| LEV | −2.096 *** |
| | (0.503) |
| ROA | 3.637 ** |
| | (1.501) |
| Constant | −5.927 *** |
| | (1.223) |
| Observations | 8358 |
| R | 0.717 |
| $R^2$ | 0.515 |

Note: CSR = corporate social responsibility. SIZE = firm size. BTM = book to market. LEV = leverage. ROA = return on assets. Standard errors in parentheses. ** $p < 0.01$, *** $p < 0.001$.

4.3.2. Board Size, Corporate Social Responsibility, and Information Asymmetry

Table 4 for Model 1 presents the results of the moderation research with BS as a moderator factor. Hypothesis **H₂ₐ** shows that board size and information asymmetry have a favourable connection. We received plenty of support for it. In the Chinese stock market, there is a strong correlation with both board size practices and, thus, the degree of asymmetric information. This connection has a positive significant influence. In other words, increasing the board size is strongly related. This implies that it is due to more information in relation to furthering BS to manage the financial press's reported issue. At the 0.01 level, the BS coefficients are positive and substantial (0.031 **), suggesting that organisations with larger boards have greater information asymmetry. The findings are consistent with past research; for instance, Hamrouni et al. (2022) and Tessema (2019) supported the statement that larger boards of directors have more difficulty coordinating and functioning and have a tendency to be less efficient in monitoring managers. This supports hypothesis **H₂ₐ**. Asymmetry in information may be impacted by the interactions between BS and CSR disclosure according to the coefficients of the cross variables CSR*BS, which are positive and significant at the 0.001 level at 0.012 ***. The connection between CSR disclosure and information asymmetry in the Chinese stock market is therefore moderated by BS, supporting our hypothesis (**H₃ₐ**). Hence, we reach the conclusion that the sensitivity of information asymmetry versus CSR disclosure is dependent on BS. In light of the fact that boards are in charge of promoting CSR disclosure and the dissemination of CSR information (Li 2008; Hamrouni et al. 2022; Jizi 2017; Adel et al. 2019), increasing the amount of information that board members disclose regarding CSR practices and strategies causes greater information asymmetry during disclosure.

**Table 4.** Test of moderation effects of CG and CSR on information asymmetry.

| Predictor | Dependent Variable—Bid–ask Spread (BAspd) | | | |
|---|---|---|---|---|
|  | Model (2) | Model (3) | Model (4) | Model (5) |
| CSR | −3.113 ** | −2.987 *** | −3.282 *** | −1.821 *** |
|  | (0.082) | (0.079) | 0.078 | (0.082) |
| BS | 0.031 ** |  |  |  |
|  | (0.012) |  |  |  |
| CSR × BS | 0.012 *** |  |  |  |
|  | (0.000) |  |  |  |
| BIND |  | −28.691 *** |  |  |
|  |  | (3.875) |  |  |
| CSR × BIND |  | 4.266 *** |  |  |
|  |  | (0.068) |  |  |
| Duality |  |  | 0.696 ** |  |
|  |  |  | (0.228) |  |
| CSR × Duality |  |  | 0.033 * |  |
|  |  |  | (0.016) |  |
| OC |  |  |  | 0.409 ** |
|  |  |  |  | (0.137) |
| CSR × OC |  |  |  | 0.013 *** |
|  |  |  |  | (0.000) |
| BTM | 0.154 ** | 0.584 *** | 0.000 *** | 0.599 *** |
|  | (0.071) | (0.111) | (0.000) | (0.128) |
| SIZE | 0.372 *** | 0.009 | −0.099 | 0.115 |
|  | (0.047) | (0.068) | (0.060) | (0.077) |
| LEV | −0.311 *** | −2.102 *** | 0.026 | −2.919 *** |
|  | (0.321) | (0.513) | (0.561) | (0.583) |
| ROA | 1.561 * | 4.112 ** | −0.318 | 5.130 ** |
|  | (0.816) | (1.529) | (0.232) | 1.746 |
| Constant | −11,123 *** | −2.217 | −1.516 | −5.516 *** |
|  | (0.776) | 0.114 | (0.191) | 1.490 |
| Observations | 8255 | 8255 | 82555 | 8255 |
| R | 0.563 | 0.705 | 0.719 | 0.589 |
| $R^2$ | 0.317 | 0.497 | 0.517 | 0.347 |

Note. CSR = corporate social responsibility. BAspd = bid–ask spread. BS = board size. BIND = proportion of independent directors. CEO Duality = when chairman and CEO are same person. OC = proportion of largest shareholder. SIZE = firm size. BTM = book to market. LEV = leverage. ROA = return on assets. Standard errors in parentheses. * $p < 0.05$, ** $p < 0.01$, *** $p < 0.001$.

### 4.3.3. Board Independence, Corporate Social Responsibility, and Asymmetry Information

The findings of the moderation research with BIND as a moderating variable are demonstrated in Table 4 for Model 2. The assumption made in the hypothesis **H₂b** that board independence has an effect (BIND) is notably negative when combined with IA. The BIND variable's coefficients are significant and negative at −28.691 *** at the 0.001 level. These findings confirm our theory by demonstrating that the BIND lessens information asymmetry **H₂b**. Our findings reveal that board independence is likely to lower agency expenses where financial service providers exercise stronger control (Fama and Jensen 1983). Therefore, a high degree of BIND is anticipated to lower capital costs for businesses with dispersed and global ownership. Additionally, the literature makes predictions regarding how the CEO will be able to affect the development of board independence. High-performing CEOs with a history of success may amass significant negotiating leverage, which they can employ to surround themselves with loyal subordinates, limiting the role of independent boards. According to the literature (Elbadry et al. 2015; Hamrouni et al. 2022; Goh et al. 2016), there is a negative correlation between board independence and the degree of asymmetric information, suggesting that independent directors contribute to lowering information asymmetry in the stock market. Nevertheless, Tessema (2019) ascertained that higher board independence is connected with increased information

asymmetry, as measured by share trading volumes and the market values of traded shares. Our findings also contradict those of Eng and Mak (2003), who argue that removing insiders from the board may be harmful to the company because outside directors may lack the expertise and experience required to correctly steer the company. The interaction between BIND and CSR disclosure influences its effectiveness in reducing asymmetric information. The CSR *BIND variable's coefficient is positive and significant at 4.266 *** at the 0.001 level. Therefore, Hypothesis $H_{3b}$ pertaining to BIND's moderating influence on CSR disclosure and asymmetric information is rejected. The interaction between BIND and the social information revealed by Chinese enterprises increases the asymmetry of market knowledge. According to the literature (Hamrouni et al. 2022), BIND and CSR can be used to reduce information asymmetry. As CSR*BIND findings are far less efficient at eliminating information asymmetry, the substantial board involvement of independent directors may substitute CSR disclosure. Put differently, when managers are less scrutinised, they are less incentivised to maximise shareholder profit and, hence, invest less in CSR (Xiang et al. 2021). Based on these findings, more independent directors on a company's board improves information transparency. Independent directors, in particular, serve as an effective corporate governance process and boost the performance of the board's oversight functions. Consequently, enhanced observation systems can lessen asymmetric information (Ahmed and Ali 2017; Elbadry et al. 2015; Hamrouni et al. 2022).

### 4.3.4. CEO Duality, Corporate Social Responsibility, and Asymmetric Information

Another facet of the board structure that has piqued the interest of researchers, according to Table 4 for Model 4, is whether the CEO also serves as chairman. The findings of this article contradict the findings of Brickley et al. (1997), who are of the opinion that the prospect of becoming the chairman is motivation for CEOs, implying that the most successful and capable CEO is likely to be appointed as chairman. Table 4 for Model 4 demonstrates the positive relationship between CEO duality and information asymmetry, with a coefficient of 0.696 ** at the 0.01 level. If the CEO has a chance to be chairman, the information will be more hidden, leading to higher levels of asymmetric information, which confirms $H_{2c}$. The most relevant prediction of this investigation is that since executive managers typically have the most detailed corporate information, companies with more significant variability in information are likely to have greater control over the CEO (Brickley et al. 1999). Consequently, it is perhaps not unexpected that Linck et al. (2008) established a substantial association between information asymmetry and the expected duties of the CEO and chairman. As a result, at the level of 0.05, the cross-variable coefficient of the SOC*CEO duality is 0.033 *. More crucially, this interaction illustrates that more CEO positions on the board increases asymmetric information, showing that $H_{3c}$ is also not acceptable. Therefore, it should be noted that the combination of CEO duality and CSR disclosure will enhance ambiguity in the Chinese stock market according to our results. Shareholders and finance professionals may not misunderstand the combining of the two systems or they may consider that such actions exceed the intended scope of corporate social responsibility and, therefore, are not financially profitable. Furthermore, this may produce increased asymmetric information on the stock market and a deterioration in the data environment.

### 4.3.5. Ownership Concentration, CSR, and Information Asymmetry

Table 4 for Model 5 confirms that ownership concentration has an extremely favourable influence on information asymmetry. This reveals that concentrated controls with a majority ownership enhance information asymmetry by 0.409 ** at the 0.01 level. This is evident from the information that the large owners will have, which will be private information, increasing information asymmetry. According to Tessema (Tessema 2019), ownership concentration is significantly connected to the level of information asymmetry, demonstrating that $H_{2d}$ is supported. The findings indicate that ownership concentration has a monitoring role in increasing information asymmetry and causing anxiety among lesser shareholders,

who suspect that management is influenced to take decisions that benefit large block holders while neglecting the smaller shareholder. We have comparable results in Column 4 when we use the SOC*OC cross-variables to capture the impact on IA. Hence, the positive interaction supporting **H$_{3d}$** continues.

*4.4. Robustness*

A second analysis was performed to investigate the robustness of the findings, investigating if the primary findings are resistant to option proxies regarding information asymmetry, as reflected in the market values of traded shares, which were incorporated from a previous study (Tessema 2019; Elbadry et al. 2015; Linsmeier et al. 2002). We repeat all of the methods from the baseline models; however, we applied the annual trading values for Equation (4) as dependent variables. The results reported in Table 5 (Models 6–10 are almost identical to the findings obtained from the baseline model estimations (1–5) shown in Tables 3 and 4. The models were then regressed to separately test each independent variable included in each model, as previously undertaken in the main results, by way of using the following equation:

$$\text{ATV}_{it} = f\left(\text{CSR}_{i,t} + \sum \text{CG}_{i,t} + \sum \text{CSR*CG}_{i,t} + \sum \text{control variables}_{i,t}\right) + \varepsilon_{it} \qquad (4)$$

Table 5 shows that Models 6–10 were regressed to determine robustness based on separate independent variables. The results supported and enhanced the primary results. The findings presented in Table 5 are relatively consistent and mainly compatible with the key findings. In Table 5, Models 6–10 demonstrate positive and substantial associations between CSR and yearly trade value, which enhances social responsibility disclosure and minimises information asymmetry between stakeholders, shareholders, and management. A greater BS reduces annual trade value and creates information asymmetry. However, an increase in the BS when revealing the CSR (CSR*BS) reduces the value of the annual trading and increases information asymmetry. Moreover, increased board independence improves yearly trade, reducing knowledge asymmetry. Interactions between board independence and corporate social responsibility (CSR*BIND) negatively and significantly influence annual trade values and increase asymmetry of information. This verifies our main findings; however, it does not support our hypothesis and differs from earlier findings. Along with the main conclusion, the data suggest that firms with greater OC may have less information than firms with lower OC, leading to higher information asymmetry and lower trading value. It is evident from this that the main findings presented in Tables 3 and 4 for Models 1–5 are supported.

**Table 5.** Alternative variable of information asymmetry.

| | Dependent Variable—Annual Trading Value (ATV) | | | | |
|---|---|---|---|---|---|
| **Predictor** | **Model (6)** | **Model (7)** | **Model (8)** | **Model (9)** | **Model (10)** |
| CSR | 0.010 ** | 0.007 * | 0.009 ** | 0.010 ** | 0.006 * |
| | (0.003) | (0.003) | (0.003) | (0.003) | (0.003) |
| BS | | −0.004 *** | | | |
| | | (0.001) | | | |
| CSR*BS | | −0.000 * | | | |
| | | (0.000) | | | |
| BIND | | | 0.263 *** | | |
| | | | (0.045) | | |
| CSR*BIND | | | −0.009 *** | | |
| | | | (0.003) | | |
| Duality | | | | −0.038 *** | |
| | | | | (0.009) | |
| CSR*Duality | | | | −0.000 | |
| | | | | (0.001) | |

**Table 5.** *Cont.*

| | Dependent Variable—Annual Trading Value (ATV) | | | | |
|---|---|---|---|---|---|
| **Predictor** | **Model (6)** | **Model (7)** | **Model (8)** | **Model (9)** | **Model (10)** |
| OC | | | | | −0.025 *** |
| | | | | | (0.005) |
| CSR*OC | | | | | −0.000 ** |
| | | | | | (0.000) |
| BTM | −0.958 *** | −0.000 *** | −0.958 *** | −0.957 *** | −0.963 *** |
| | (0.004) | (0.000) | (0.004) | (0.004) | (0.004) |
| SIZE | 0.941 *** | 0.949 *** | 0.940 *** | 0.939 *** | 0.939 *** |
| | (0.004) | (0.003) | 0.003 | (0.003) | (0.003) |
| LEV | −1.998 *** | −1.980 *** | −1.996 *** | −2.010 *** | −1.997 *** |
| | (0.020) | (0.020) | (0.020) | (0.020) | (0.20) |
| ROA | 0.493 *** | 0.491 *** | 0.466 *** | 0.494 *** | 0.463 *** |
| | (0.59) | (0.58) | (0.059) | (0.059) | (0.059) |
| Constant | 1.198 *** | 0.865 *** | 1.143 *** | 1.229 *** | 1.274 *** |
| | (0.048) | (0.054) | (0.049) | (0.049) | (0.051) |
| Observations | 8117 | 8117 | 8117 | 8117 | 8117 |
| R | 0.988 | 0.989 | 0.988 | 0.988 | 0.988 |
| $R^2$ | 0.977 | 0.977 | 0.977 | 0.977 | 0.977 |

Note: CSR = corporate social responsibility. ATV = annual trading value. BS = board size. BIND = percentage of independent directors. CEO Duality = when chairman and CEO are same person. OC = proportion of largest shareholder. SIZE = firm size. BTM = book to market. LEV = leverage. ROA = return on assets. Standard errors in parentheses. * $p < 0.05$, ** $p < 0.01$, *** $p < 0.001$.

In Figure 2, we can notice that CSR disclosure reduces the information asymmetry between companies and investors. Furthermore, the results of the study suggest that several aspects of CG play a moderating role when it comes to the relationship between CSR and information asymmetry. More specifically, board size and CEO duality, as well as board independence positively affect the bid–ask spread. CSR disclosure and information asymmetry are positively influenced by moderation via board independence.

## 5. Conclusions, Limitations, and Future Research

The current study investigated the function of corporate governance features, specifically board composition, board activity, and ownership concentration as mediators, in addition to their impact on the relationship between corporate social responsibility and information asymmetry in Chinese listed companies. Although several earlier studies assessed the direct influence of the disclosure of corporate social responsibility on the information environment, there is a scarcity of research that focuses on the interaction between corporate social responsibility disclosure and various governance features. Previous research, however, indicates that the board of directors has a crucial role in determining company reporting policies. Based on previous research, we suggest that corporate governance interactions with CSR disclosure can act as an alternative and supplement minimising information asymmetry (Cormier et al. 2011; Hamrouni et al. 2022).

The first hypothesis, that CSR disclosure negatively affects information asymmetry (IA), is validated by the results shown in Tables 3–5. This is in agreement with all of the coefficients. Furthermore, evidence suggests that optimal practices in corporate governance are inversely associated with knowledge asymmetry. The findings support our predictions regarding the effectiveness of governance tools in improving the market by reducing the bid–ask spread and increasing share trading volume. Shareholders are more likely to invest when a business follows best practices in corporate governance because they are more certain that agency concerns will be controlled. According to the results, the level of information asymmetry rises as the number of board members increases, whilst increasing the number of independent directors reduces that imbalance and boosts trading value. Board independence minimises information asymmetry according to the literature

(Hillier and McClgan 2006; Kanagaretnam et al. 2007; Holm and Schøler 2010). Furthermore, it is congruent with the concerns highlighted by Jensen and Meckling (1976), namely that the absence of independent leadership makes it impossible for the board to respond effectively when the senior management team fails. As a consequence, board independence has a substantial impact on information asymmetry. The findings demonstrate that the CSR*BIND test regression was positively associated with the relationship between CSR and AI, implying that BIND is operating as a surrogate for CSR. These outcomes are equivalent to those obtained by Hamrouni et al. (2022).

Additionally, there was a substantial supportive association between ownership concentration and information asymmetry, indicating that ownership intensity increases information asymmetry. This is in contrast to the literature (Pawlina and Renneboog 2005; Shleifer and Vishny 1997; Florackis and Ozkan 2009; Perotti and von Thadden 2003), which determined that the majority shareholders can reduce asymmetry information and increase long-term effectiveness. However, the results are comparable with those previously published by Rehman et al. (2022). Likewise, the regulated role of OC in CSR (CSR*OC) is related to CSR and IA. This contradicts the findings obtained by Rehman et al. (2022), which revealed a significant adverse correlation between CSR*OC and IA. Furthermore, the robust test for information asymmetry using an alternative variable reinforced the results. However, the scope of this study is limited because it focuses on non-financial enterprises in China. Furthermore, because of the significant amount of missing data during the investigation, additional indicators of information asymmetry were not taken into consideration in this analysis. Despite this, our research has enhanced the literature in a variety of ways. Firstly, this study sheds light on how board and ownership concentration affects information asymmetry, which differs depending on the economic, legal, and political institutions, and regulatory incentives, in addition to the social environment. Moreover, the incorporation of corporate governance as a moderating variable spanning corporate social responsibility and asymmetry information helps us to acquire a better understanding of what is taking place behind the scenes.

However, despite its possible contributions and the additional insight of this study, the findings of the current research are subject to a number of limitations that could be discussed in future research. The most noticeable limitation is that our study applies to only large, public Chinese organisations. Studies on other sorts of firms, such as SMEs, are needed for future studies. Furthermore, future research could examine the association between CSR and sustainability. Reporting this can contribute to the literature from a different viewpoint. This research emphasises various aspects of governance mechanisms, such as the board of directors; however, future research may explore other forms of external corporate governance, for instance, institutional ownership. Therefore, future studies could use continuous variables. Likewise, further studies can also examine the moderating role of different forms of CG. Employing various measurements for CG would enhance the degree of CG control. One proxy for measuring ownership structure may be lacking in relation to drawing a certain assumption concerning the role of ownership.

Additional research requires measuring the effect of CSR on financial performance in different countries. A study on why and how the commitment of companies to CSR varies among countries might offer further insight into the complicated relationship between CSR and CG. Similarly, future studies must also investigate the circumstantial factors that create a moral judgment and ethical interpretation of CSR within different cultures. Notwithstanding these shortcomings, our conclusion provides an organisational exercise by presenting data on the fundamental role of CG. One possible drawback of the present research is the application of measures for IA, since this research also employed bid–ask spread as a proxy for IA, which is a popular instrument employed in previous studies. It would be interesting if future studies validated the results of this study by using various other measures.

A future study could employ another proxy for measuring information asymmetry, stock return volatility, and the probability of informed trading score (PIN) (Mohamed and

Rashed 2021). An additional drawback of our research is that our analysis is restricted to a sample of Chinese listed companies. Future studies may explore and compare the variations between different sectors in terms of CSR practices in developing countries. Different measurements are employed to measure CSR disclosure. Moreover, future studies should reflect the institutional environment of China. According to the previous limitations, the research can be performed in various institutional contexts. Future studies might also consider different moderating variables that impact CSR, such as the roles of the CEO and audit committee, in addition to internal controls and media exposure in the current digitalisation era. Likewise, addressing the effects of these additional proxies that may affect the relationship between CSR and information asymmetry should be investigated and could offer additional insight.

**Author Contributions:** Conceptualization, F.A. and N.A.A.; Data curation, F.A.; Formal analysis, F.A.; Funding acquisition, F.A., N.A.A. and R.A.; Investigation, F.A. and N.A.A.; Methodology, F.A. and N.A.A.; Project administration, F.A.; Software, F.A.; Supervision, F.A.; Validation, F.A. and N.A.A.; Visualization, F.A. and N.A.A.; Writing—original draft, F.A., N.A.A. and R.A.; Writing—review and editing, F.A., N.A.A. and R.A. All authors have read and agreed to the published version of the manuscript.

**Funding:** This research received no external funding.

**Institutional Review Board Statement:** Not applicable.

**Informed Consent Statement:** Not applicable.

**Data Availability Statement:** The data presented in this study are available on request from the first author.

**Acknowledgments:** The publication of this research has been supported by the Deanship of Scientific Research and Graduate Studies at Philadelphia University, Jordan.

**Conflicts of Interest:** The authors declare no conflict of interest.

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
