# Peer review of "The Moderating Effect of Corporate Governance on Corporate Social Responsibility and Information Asymmetry: An Empirical Study of Chinese Listed Companies"

_economies, doi:10.3390/economies10110280_

Round 1
Reviewer 1 Report
See the attached file

Author Response
Reviewer 1
Comments of “ The moderating effect of corporate governance on corporate social responsibility and information asymmetry: An empirical study of Chinese listed companies” [Economies] Manuscript ID: economies-1930099
The title of paper is interesting but the content needs to be revised and restructured. The moderating effect of corporate governance (CG) on corporate social responsibility (CSR) and information asymmetry (IA) is a good research question but the content of this paper is not clear for readers to follow up. Comments and suggestions are as follows.
- The current version of Abstract is too long. For example, in Line 20-21, some control variables not included does not need to be shown here. What is IA in abbreviation (Line 28)? In Line 31-33, that is not a complete sentence. In Line 3334, is “CSR promote transparency between managers and stakeholder groups” the main purpose of this study?
Thanks for your comments. We have made the necessary changes and shortened the abstract.
- Line 39 (China, practice CSR…..), Line 45 (…, To…..), Line 58 (courage Supervisors….), Line 60 (…IA….) etc. Are they typos?
Thanks for your comments. Yes , we have made changes.
- In Line 88, Secondly, …… One sentence for the second contribution is too short and not clear.
Thanks for your comments. We have made change.
- In Line 147, “2.2 CG as well as asymmetry information” is not easy to understand the subtitle.
Thanks for your comments. We have made change.
- In Line 358-359, where is the definition of this variable (annual trading value) in the equation model?
Thanks for your comments. The definition of the variable annual trading value (ATV) in the equation3 and table 5.
- Please read the content again and make sure your research questions and research design clear and easy for readers to follow up.
Thank you for your comments, we will read and make proofreading for it.
Reviewer 2 Report
Abstract: it should be rearranged in a more rational way (I suggest: Purpose Design/Methodology/Findings/Practical Implications/Originality and value).
Introduction: in line 38 and line 67 you affirm two opposite think; especially line 38 seems not very shareable, therefore it should be better argued and supported by data; line 46 "to put it another way" seems redundant and the following "it" refers to what; if you use acronymous, you should identify them in the text; line 71 what report are you referring to?
Conclusion: it is not supposed to be just a summary of the results exposed in the previous paragraph, but it should contain a discussion by the authors, limitations and future research.
Spelling: many words are in capitol letter, apparently, whiteout any reason.
Author Response
Reviewer2
Open Review
( ) I would not like to sign my review report
(x) I would like to sign my review report
English language and style
( ) Extensive editing of English language and style required
(x) Moderate English changes required
( ) English language and style are fine/minor spell check required
( ) I don't feel qualified to judge about the English language and style
|
Yes |
Can be improved |
Must be improved |
Not applicable |
|
|
Does the introduction provide sufficient background and include all relevant references? |
( ) |
( ) |
(x) |
( ) |
|
Are all the cited references relevant to the research? |
(x) |
( ) |
( ) |
( ) |
|
Is the research design appropriate? |
(x) |
( ) |
( ) |
( ) |
|
Are the methods adequately described? |
(x) |
( ) |
( ) |
( ) |
|
Are the results clearly presented? |
(x) |
( ) |
( ) |
( ) |
|
Are the conclusions supported by the results? |
( ) |
(x) |
( ) |
( ) |
Comments and Suggestions for Authors
Abstract: it should be rearranged in a more rational way (I suggest: Purpose Design/Methodology/Findings/Practical Implications/Originality and value).
Thanks for your comments. We have made the necessary changes and shortened the abstract based your comments.
Introduction: in line 38 and line 67 you affirm two opposite think; especially line 38 seems not very shareable, therefore it should be better argued and supported by data; line 46 "to put it another way" seems redundant and the following "it" refers to what; if you use acronymous, you should identify them in the text; line 71 what report are you referring to?
Thanks for your comments. We have made the necessary changes based your comments.
Conclusion: it is not supposed to be just a summary of the results exposed in the previous paragraph, but it should contain a discussion by the authors, limitations and future research.
Thanks for your comments. We have made the necessary changes based your comments and add some paragraphs for limitations and future research.
Spelling: many words are in capitol letter, apparently, whiteout any reason.
Thanks for your comments. We made a proofreading.

Reviewer 3 Report
The paper studies the function of corporate governance features (board composition, board activity, and ownership concentration) as mediators and their impact on the relationship between CSR and IA in Chinese listed companies.
I think the topic is interesting but some revisions are needed.
1. The research background must be clarified. Particularly in the abstract eand the introduction the authors should explain bettere their focus, the specific research question they wish to answer to.
2. Besides, exposition is not always clear. Some key concepts need to be introduced at the beginning in order to help the readers to understand immediately the rationale of the paper.
3. English language must be improved as well as typing errors must be avoided
4. Implications of the study should be enhanced (they shoul be included also in the abstract)
Author Response
Reviewer3
Open Review
( ) I would not like to sign my review report
(x) I would like to sign my review report
English language and style
(x) Extensive editing of English language and style required
( ) Moderate English changes required
( ) English language and style are fine/minor spell check required
( ) I don't feel qualified to judge about the English language and style
|
Yes |
Can be improved |
Must be improved |
Not applicable |
|
|
Does the introduction provide sufficient background and include all relevant references? |
( ) |
( ) |
(x) |
( ) |
|
Are all the cited references relevant to the research? |
(x) |
( ) |
( ) |
( ) |
|
Is the research design appropriate? |
( ) |
( ) |
(x) |
( ) |
|
Are the methods adequately described? |
(x) |
( ) |
( ) |
( ) |
|
Are the results clearly presented? |
(x) |
( ) |
( ) |
( ) |
|
Are the conclusions supported by the results? |
( ) |
(x) |
( ) |
( ) |
Comments and Suggestions for Authors
The paper studies the function of corporate governance features (board composition, board activity, and ownership concentration) as mediators and their impact on the relationship between CSR and IA in Chinese listed companies.
I think the topic is interesting but some revisions are needed.
- The research background must be clarified. Particularly in the abstract eand the introduction the authors should explain bettere their focus, the specific research question they wish to answer to.
Thanks for your comments. We have made the necessary changes based your comments.
- Besides, exposition is not always clear. Some key concepts need to be introduced at the beginning in order to help the readers to understand immediately the rationale of the paper.
Thanks for your comments. We have made the necessary changes based your comments.
- English language must be improved as well as typing errors must be avoided
Thanks for your comments. We made a proofreading.
- Implications of the study should be enhanced (they shoul be included also in the abstract)
Thanks for your comments. We have made the necessary changes based your comments and add some paragraphs for implications.
Round 2
Reviewer 3 Report
Authors have changed and imprved the earlier version of the paper
I think now it is fine to be published